# Comparison of Cinchona Catalysts Containing Ethyl or Vinyl or Ethynyl Group at Their Quinuclidine Ring

**DOI:** 10.3390/ma12183034

**Published:** 2019-09-18

**Authors:** Sándor Nagy, Zsuzsanna Fehér, Gergő Dargó, Júlia Barabás, Zsófia Garádi, Béla Mátravölgyi, Péter Kisszékelyi, Gyula Dargó, Péter Huszthy, Tibor Höltzl, György Tibor Balogh, József Kupai

**Affiliations:** 1Department of Organic Chemistry & Technology, Budapest University of Technology & Economics, Szent Gellért tér 4, H-1111 Budapest, Hungary; 2Chemical Department, Chemical Works of Gedeon Richter Plc., P.O. Box 27, H-1103 Budapest, Hungary; gytbalogh@mail.bme.hu; 3Department of Inorganic & Analytical Chemistry, Budapest University of Technology & Economics, Szent Gellért tér 4, H-1111 Budapest, Hungary; 4Furukawa Electric Institute of Technology, Késmárk utca 28/A, H-1158 Budapest, Hungary; 5Department of Chemical & Environmental Process Engineering, Budapest University of Technology and Economics, Műegyetem rkp. 3, H-1111 Budapest, Hungary

**Keywords:** organocatalysis, cinchona, squaramide, thiourea, p*K*_a_, Michael addition

## Abstract

Numerous cinchona organocatalysts with different substituents at their quinuclidine unit have been described and tested, but the effect of those saturation has not been examined before. This work presents the synthesis of four widely used cinchona-based organocatalyst classes (hydroxy, amino, squaramide, and thiourea) with different saturation on the quinuclidine unit (ethyl, vinyl, ethynyl) started from quinine, the most easily available cinchona derivative. Big differences were found in basicity of the quinuclidine unit by measuring the p*K*_a_ values of twelve catalysts in six solvents. The effect of differences was examined by testing the catalysts in Michael addition reaction of pentane-2,4-dione to *trans*-β-nitrostyrene. The 1.6–1.7 p*K*_a_ deviation in basicity of the quinuclidine unit did not result in significant differences in yields and enantiomeric excesses. Quantum chemical calculations confirmed that the ethyl, ethynyl, and vinyl substituents affect the acid-base properties of the cinchona-thiourea catalysts only slightly, and the most active neutral thione forms are the most stable tautomers in all cases. Due to the fact that cinchonas with differently saturated quinuclidine substituents have similar catalytic activity in asymmetric Michael addition application of quinine-based catalysts is recommended. Its vinyl group allows further modifications, for instance, recycling the catalyst by immobilization.

## 1. Introduction

Asymmetric syntheses went through explosive growth in the last decades [1,2,3,4,5]. After transition metal- and enzyme-based catalysis, the application of organocatalysis has gained ground in the field of asymmetric syntheses. The aim of enantioselective organocatalytic synthesis is to produce enantiopure compounds from achiral substrates facilitated by asymmetric organocatalysts. The importance of the synthesis and application of enantiopure products lies in the fact that many industries are required to produce the appropriate enantiomer, for instance as drug or pesticide intermediates.

Within organocatalysis, bifunctional catalysts have become widespread. This bifunctionality means the capability to activate two components of a reaction simultaneously [6]. Cinchona moiety is one of the privileged chiral skeletons in asymmetric organocatalysis [7]. Its structure consists of two rigid rings, the aromatic quinoline, and the alicyclic quinuclidine rings. The tertiary amino group in the quinuclidine ring gives basic character for the molecule, hence this can activate/fix nucleophile or electrophile, and possessing a chiral skeleton, cinchona unit is also responsible for chiral induction.

There are many successful applications of cinchona-based organocatalysts in asymmetric reactions with high yields and enantiomeric excesses [8]. Such reactions include, but are not limited to, Morita–Baylis–Hillman reaction, Henry reaction, Diels–Alder reaction, and the most often examined, Michael addition. These reactions are used for the synthesis of drug intermediates, such as oseltamivir, baclofen, or prostaglandin E1 [9,10,11].

When bifunctional H-bond organocatalysts, such as cinchona-thioureas or squaramides were applied in Michael additions of 1,3-dioxo compounds to nitroalkenes, it was found [12,13,14,15,16] that the reaction can proceed by two mechanisms. In one of these mechanisms, the acidic NH groups (of thiourea or squaramide moieties) activate the enolate, while the protonated quinuclidine fixes the electrophile (Figure 1a). According to the other mechanism, the activation of substrates occurs on the opposite sites of catalysts (Figure 1b). Nevertheless, both mechanisms start with the deprotonation of the nucleophile by the basic N atom of the quinuclidine ring.

Based on these mechanisms, the basic N atom of the quinuclidine unit plays an important role in these reactions, which is related to its p*K*_a_ value. These cinchona-based organocatalysts are usually used with saturated or unsaturated substituent (ethyl, vinyl, ethynyl) on the quinuclidine ring, although the effect of that saturation has not been examined before. Vinyl or ethynyl group is often the target for further transformations (Figure 2) such as polymerization or immobilization [17,18,19,20,21,22]; however, several methods for modification are described in the literature in other positions of the cinchonas as well (C9 position, or modification after demethylation of quinoline) [23,24].

Herein we report the study of substituent influence on the basicity of cinchonas. We synthesized the appropriate hydroxyl, amino, thiourea, and squaramide cinchona derivatives with different saturation of the quinuclidine substituent. We measured the p*K*_a_ values of these derivatives in eight solvents and examined the catalytic activity of these twelve catalysts in Michael addition of pentane-2,4-dione to *trans*-β-nitrostyrene in seven solvents.

## 2. Results and Discussion

Since the strength of hydrogen bonding interactions plays an important role in H-bond organocatalysis [16,25,26,27], determination of p*K*_a_ values could help understanding the mechanisms and catalytic activity, therefore, it can contribute the design of more efficient catalytic systems. In cases when deprotonation and/or hydrogen bonding are able to promote reactions or could increase enantioselectivity, knowing the p*K*_a_ values of the corresponding groups is essential.

### 2.1. The pK_a_ Values of Conjugate Acid Forms of Quinuclidine

To get further insight into the H-bond organocatalytic behavior of the aforementioned twelve catalysts, the p*K*_a_ values of all four compound classes were measured in six different solvents by UV-spectrophotometric titrations (Pion Inc., Forest Row, UK) (Table 1 and Appendix A). For better clarity, only those p*K*_a_ values are shown in Table 1 which were measured in water for the conjugate acid forms of basic quinuclidine.

### 2.2. The Effect of the Substituents on Quinuclidine Basicity

Within the four different compound classes (cinchona hydroxyl, amino, thiourea, and squaramide derivatives) the effect of the substituents of quinuclidine on the basicity was examined. As expected, the saturation of the substituent leads to increased basicity due to electronic reasons. The difference among the p*K*_a_ values of the tertiary amino nitrogen in quinuclidine is in a range of 1.6–1.7 in each compound class, depending on the substituents.

### 2.3. The Effect of the C9 Group of Cinchona on Quinuclidine Basicity

Not only the aliphatic substituent of quinuclidine unit has an effect on the basicity, but the groups of C9 position of the cinchona as well, since these groups and the aliphatic substituents are practically equidistant from the quinuclidine nitrogen. The decreasing order in basicity is the following: amino, hydroxyl, thiourea, and squaramide. This may be caused mainly by electronic effects. Thiourea derivatives have higher basicity than that of the corresponding squaramides. This may be explained by the squaramide moiety having aromatic feature, therefore the electron-withdrawing effect of the bis(trifluoromethyl)phenyl group can have a stronger effect than in the case of thiourea.

### 2.4. Relationship Between pK_a_ Values and H-Bonding Abilities

Based on our published results [28,29,30] and the relevant literature [31,32], in asymmetric Michael additions of 1,3-dioxo compounds to β-nitroalkenes-cinchona, squaramides or thioureas are recommended to be used as enantioselective organocatalysts. Owing to their H-bonding ability (quantified by their p*K*_a_ values) [33,34,35,36,37,38] and chiral skeleton, they not only give high yields but excellent enantioselectivities as well. Despite the fact that the N atom of the quinuclidine ring in thioureas is more basic, the observed enantiomeric excesses are similar than those given by squaramides. This observation can be explained by the different strength of H-bonds formed by thioureas and squaramides.

According to the mechanisms [12,13,14], cinchona thiourea or squaramide can fix substrates by three H-bonds, while hydroxyl or amino derivatives are able to interact different ways, for instance with less H-bonds. This causes the notable differences among the enantioselectivities. The strong H-bonds can be explained by the relatively high acidity of these units. The p*K*_a_ values of thiourea and squaramide units measured in water can be seen in Table 2 (in other solvents see Appendix A).

Based on p*K*_a_ values, squaramides are more acidic than thioureas, hence stronger H-bonds can be formed between this moiety and the corresponding substrate. Taking into account the bifunctionality of cinchona thioureas and squaramides, they possess similar catalytic activity: while the quinuclidine nitrogen of cinchona thioureas are more basic, squaramides acidic NH moiety can form stronger H-bonds with the substrate.

Reflecting to further importance of the acidity of the H-bond donor units, an analogue of squaramides, the thiosquaramides having more acidic NH groups should be mentioned [30,38,39,40,41,42]. Thiosquaramides perform better results (yield and ee) in asymmetric Michael addition than the corresponding squaramides. Thiosquaramides could act as Brønsted acids, too. Therefore, they can also catalyze aza-Diels–Alder reaction [30,42], in which neither thioureas nor squaramides can give the desired product.

### 2.5. Synthesis of the Catalysts

We synthesized the cinchona catalysts starting from the commercially available quinine (**Q**), which was transformed to hydroquinine (**HQ**) by catalytic hydrogenation (Scheme 1). Didehydroquinine (**DQ**) was prepared in a two-step reaction from quinine: a bromine addition was followed by HBr elimination [43].

Cinchona amino derivatives (**HQ-N**, **Q-N**, **DQ-N**) were prepared starting from the relevant hydroxy derivatives (**HQ**, **Q**, **DQ**) based on previously reported procedures [44]. In the next step, thiourea derivatives (**HQ-TU**, **Q-TU**, **DQ-TU**) were obtained by the addition of the appropriate amine derivatives to bis(trifluoromethyl)phenyl isothiocyanate (**SCN**) [45,46]. Finally, addition of the abovementioned amines to half-squaramide methyl ester (**HSQ**) resulted in the corresponding squaramides (**HQ-SQ**, **Q-SQ**, **DQ-SQ**) [46,47]. During the synthetic procedures, all compounds were characterized by well-established methods including MS, IR, ^1^H, and ^13^C NMR spectroscopies.

### 2.6. Application of the Catalysts in Asymmetric Michael Addition

We used hydroxy, amine, thiourea, and squaramide type catalysts in Michael addition (Scheme 2) of pentane-2,4-dione (**2**) to *trans*-β-nitrostyrene (**1**). Conventional and green solvents including dichloromethane (DCM), methyl *tert*-butyl ether (MTBE), toluene, acetonitrile (MeCN), ethyl acetate (EtOAc), methanol, and tetrahydrofuran (THF) were used (Table 3 and Appendix A) under the same conditions. In methanol, which is a protic solvent, we got only lower yields (up to 82% vs. up to 100%) and enantioselectivities (up to 51% ee vs. up to 93% ee). This observation is in agreement with similar cinchona-based organocatalysis [48]. In methanol, hydrogen bonds could form between the solvent and the catalysts/reactants. Accordingly, to apply a hydrogen bonding organocatalyst in protic polar solvents, could result in low yields and enantiomeric excesses.

Michael adducts were obtained in high yields (up to 99%) when hydroxyl or amino derivatives of cinchona were applied, but these reactions gave only 34% ee (see Appendix A). Concerning selectivity, no significant difference was found between the differently saturated cinchonas; however, among hydroxy derivatives in most of the solvents, **HQ** gave higher enantiomeric excesses. It can be explained by its higher basicity, but this tendency cannot be observed in cases of other derivatives.

The utilization of squaramides or thioureas served both high yields (up to 100%, see Table 3) and enantioselectivities (up to 93% ee, see Table 3 and Figure 3). The different p*K*_a_ values of these derivatives did not lead to significant changes in yield or enantiomeric excess. Squaramides and thioureas gave good results in asymmetric Michael addition, without showing any considerable difference in the catalytic activity.

### 2.7. Theoretical Calculations

Due to the importance of hydrogen bonds and acidity in the catalytic mechanisms of cinchona-thiourea compounds [49,50], one of the main questions is how the ethyl, ethynyl, and vinyl substituents can affect their acid-base properties. Therefore, we investigated systematically the different tautomers of the synthesized **HQ-TU**, **Q-TU**, and **DQ-TU** catalysts and their protonated and deprotonated forms using quantum chemical calculations.

In agreement with their similar reactivities observed in the experiments, the geometric structures of the neutral cinchona and thiourea frameworks (see Appendix A) and also the energetics of the different tautomers, as well as the protonation and deprotonation Gibbs free energies (Figure 4 and Appendix A, Table 4 and Appendix A) depend only slightly on the ethyl, ethynyl, or vinyl substituents. In all cases, the most stable tautomer is the thione form, containing two NH groups (*i* on Figure 4), while the quinuclidine nitrogen is protonated in the higher-lying tautomers *ii* and *iii*. Interestingly, in the higher-lying isomers the proton detaches from the bis(trifluoromethyl)phenyl side of the thiosquaramide. We expect that this is due to the interaction with the nitrogen lone pair with the π system, leading to an increased acidity at this site.

The thiol form *iv* is lying even higher in free energy. Thus, overall the calculations show that among the possible tautomers the catalytically most active [49,50] *i* is the most stable form and present in considerably higher amount than the others. 

Electrostatic potential provides important insights into the intermolecular interactions of similar compounds [51,52]. The electrostatic potentials (see the Appendix A) clearly show that both the sulfur and the nitrogen atoms are possible protonation sites. The calculations showed that in line with the above discussion, the quinuclidine nitrogen is the preferred site in all cases, with proton affinity of ~−44 kJ·mol^−1^ (Figure 4). The protonated thiol form of the cinchona-thiourea lies about 62 kJ·mol^−1^ higher in free energy, while the proton affinity decreases to −14 kJ·mol^−1^. Thus, the computations show that the quinuclidine is the main protonation site in acidic conditions.

Interestingly, in contrary to the neutral tautomers, there is only 9–12 kJ·mol^−1^ difference between the two possible deprotonated forms, indicating that both of the possible tautomers present in acidic conditions.

## 3. Conclusions

In conclusion, we can approach bifunctional H-bond organocatalysts in two ways: focusing on their H-bond acceptor or donor abilities (Figure 5).

As H-bond acceptors: within each compound class (hydroxyl, amino, squaramide, thiourea) the bifunctional cinchona organocatalysts with differently saturated substituents on basic quinuclidine nitrogen have 1.6–1.7 difference in their p*K*_a_ values. The strongest bases are the compounds substituted with ethyl, followed by vinyl and then ethynyl groups. No significant difference was found in yields and enantioselectivities in any compound classes applying them in Michael addition. The basicity in each compound class increases in the following order: amino, hydroxyl, squaramides, thiourea.

As H-bond donors: the catalysts, which are able to form dual H-bonds gave the highest enantiomeric excesses (thioureas and squaramides). The squaramides have by 1.3–1.6 lower p*K*_a_ values than the corresponding thioureas, resulting in stronger H-bonds with the substrate.

By summarizing the H-bond abilities, cinchona thioureas are better acceptors, but as donors, squaramides show stronger interactions. As organocatalysts in Michael addition, there is no considerable difference among them, both the yields and enantiomeric excesses were similar in each solvent and, within these compounds, the saturation had no effect on yield and ee as well.

Based on our results, saturation of the substituent at the quinuclidine unit has no significant effect on the catalysis thus the most easily available quinine derivatives (vinyl substituent at quinuclidine) are worth to apply as cinchona-based organocatalysts as themselves or as starting materials of their C9 derivatives. Considering immobilization, vinyl is a suitable group for further modifications, for instance in polymerization, or forming a reactive group in the molecule for this purpose.

Our results were corroborated by computational studies. Accordingly, the saturation of the substituent on quinuclidine has a slight effect on basicity of the H-bond acceptor nitrogen atom. Moreover, we accomplished calculations on stability of different tautomer forms of thiourea catalysts. These calculations approved that the most stable tautomers in all cases are the thione forms with the proton residing on the nitrogen atom of the quinuclidine.

## 4. Experimental

The UV/pH titrations were performed using D-PAS technique (Dip-Probe Absorbance Spectrophotometry, a quartz fiber dip probe measuring absorbance using a flow-cell) attached to a SiriusT3 instrument (Pion Inc., Forest Row, UK) [53,54]. The p*K_a_* values were calculated by SiriusT3Refine™ software (Pion Inc., Forest Row, UK). All measurements were performed in solutions of 0.15M KCl under nitrogen atmosphere, at *t* = 25.0 ± 0.1 °C. The cosolvent dissociation constants (p_s_*K*_a_ values) of the compounds were also determined in various MeOH–water mixtures between 15 and 70 wt% and MeCN–water, dioxane–water, THF–water, DMSO–water mixtures between 15 and 50 wt%. Each sample was measured at least in minimum six different cosolvent–water ratios. To obtain the 98.0% and 99.9% cosolvent p_s_*K*_a_ values from p_s_*K*_a_ data, linear and Yasuda–Shedlovsky extrapolation methods have been used. The Yasuda–Shedlovsky method establishes a correlation with the dielectric constant and uses the following equation: p_s_*K*_a_ + log[H_2_O] = *a ε* + *b*, where log[H_2_O] is the molar water concentration of the given solvent mixture. This method is the most widely used procedure in cosolvent techniques [55,56].

Infrared spectra were recorded on a Bruker Alpha-T FT-IR spectrometer (Bruker, Ettlingen, Germany). Optical rotations were measured on a Perkin-Elmer 241 polarimeter (Perkin-Elmer, Waltham, MA, USA) that was calibrated by measuring the optical rotations of both enantiomers of menthol. NMR spectra were recorded at Directorate of Drug Substance Development, Egis Pharmaceuticals Plc., on a Bruker Avance III HD (Bruker, Ettlingen, Germany) (at 600 MHz for ^1^H and at 150 MHz for ^13^C spectra). Mass spectra were recorded on CAMAG LCMS Interface (CAMAG, Muttenz, Switzerland) (HPLC pump: Shimadzu LC-20AD Prominence SQ MS: Shimadzu LCMS-2020 MS settings: detector voltage: 1.10 kV, *m/z*: 105–1000, scan speed: 1075 u·s^−1^, DL temperature: 250 °C, nebulizing gas flow: 1.5 L·min^−1^, drying gas flow: 15 L·min^−1^, eluent: acetonitrile: 0.1% (*v/v*) formic acid 95:5, 1.5 mL·min^−1^). The exact mass measurements were performed using Q-TOF Premier mass spectrometer (Waters Corporation, 34 Maple St, Milford, MA, USA) in positive electrospray ionization mode. The enantiomeric ratios of the samples were determined by chiral high-performance liquid chromatography (HPLC) measurements using reversed-phase mode (Thermo Finnigan Surveyor LC System, Thermo Fisher Scientific, Waltham, MA, USA). Elemental analyses were performed in the Microanalytical Laboratory of the Department of Organic Chemistry, Institute for Chemistry, Eötvös Loránd University, Budapest, Hungary. Melting points were taken on a Boetius micro-melting point apparatus (VEB Dresden Analytik, Dresden, Germany) and they were uncorrected. Starting materials were purchased from Aldrich Chemical Company (St. Louis, MO, USA) unless otherwise noted. Silica gel 60 F_254_ (Merck, Darmstadt, Germany) plates and aluminium oxide 60 F_254_ (Merck) were used for TLC. The spots of materials on TLC plates were visualized by UV light at 254 nm. Silica gel 60 (70–230 mesh, Merck) was used for column chromatography. Ratios of solvents for the eluents are given in milliliters.

The lowest energy tautomers were obtained using the MMFF94 force, as it is implemented in the Avogadro program [57]. Subsequently, the geometry optimization of the neutral structures as well as of their protonated and deprotonated forms were carried out using Density Functional Theory applying the ωB97X-D functional [58] and 6-31G* basis set [59], as it is implemented in the Q-Chem 5.2 quantum chemical software package [60], while the final energy computations were performed using the 6-311++G** basis set. The applied density functional includes long range and dispersion corrections and the accuracy of this method has been tested for similar systems in our previous studies [28,29,30]. The (75,302) integration grid was applied in all cases. The geometries of the catalysts were optimized both in the gas phase and in THF solvent using SM8 [61] and the SS(V)PE [62]. Harmonic vibrational frequencies were computed in the gas phase for the most stable tautomers. These show no imaginary vibrational frequencies, confirming that the computed conformers are minima on the potential energy surface. Single-point calculations were carried out using the SM12 solvent model [63]. As it is expected, the different solvent models yield similar results for the relative energies of the different structures (the maximum absolute deviation is 14 kJ mol^−1^, see Appendix A) and similar accuracy is obtained for protonation free energies (the maximum absolute error is 13 kJ mol^−1^), however much larger differences are obtained for anions (maximum error in deprotonation free energies is 47 kJ mol^−1^). Nevertheless, both solvent models yield the same tendencies. During the computation of the relative Gibbs free energies, the vibrational contribution is neglected and only the solvation contribution is considered. For the protonation and deprotonation computation, the solvation Gibbs free energy of the proton in THF is necessary, however, its computation is a tedious task. We approximated this quantity using a supermolecule model consisting of a proton and two THF molecules, embedded in a continuum solvent [64]. The molecules were visualized using the PyMol program [65].

### 4.1. Hydroquinine

A solution of quinine (2.00 g, 6.17 mmol) in methanol (20 mL) was added to a suspension of Pd/C in methanol (20 mL) under argon atmosphere. The quinine was hydrogenated at 25 °C under atmospheric pressure. When the reaction was completed (1 h), the reaction mixture was filtered through a pad of Celite. The solvent was evaporated, giving pure hydroquinine (**HQ**, Figure 6) as a white solid (2.01 g, 99%). Spectral data are fully consistent with those reported in the literature [66].

### 4.2. (S)-((1S,2S,4S,5S)-5-Ethynylquinuclidin-2-yl)(6-methoxyquinolin-4-yl)methanamine

Triphenylphosphine (976 mg, 3.72 mmol) was added to a solution of **DQ** (1.00 g, 3.10 mmol) in DCM (6 mL) under argon atmosphere and stirred for 1 h at room temperature. The reaction mixture was cooled down to 0 °C and a solution of DIAD (740 µL, 753 mg, 3.72 mmol) in DCM (1 mL) was added to it dropwise. After addition, the reaction mixture was stirred at room temperature overnight. Then the reaction mixture was cooled down to 0 °C, and water (1.35 mL, 75 mmol) was added to it in one portion. After this step, a solution of triphenylphosphine (1.46 g, 5.58 mmol) in DCM (5 mL) was added dropwise to the reaction mixture and it was stirred overnight at 25 °C. After the reaction was completed, the mixture was diluted to 50 mL with DCM, and it was shaken with an aqueous hydrochloric acid solution (2 × 50 mL, 10 w%). The combined inorganic phase was cooled down to 0 °C, and its pH was adjusted to 10 with aqueous sodium hydroxide solution and extracted with DCM (2 × 50 mL). The combined organic phase was dried over MgSO_4_, and the solvent was evaporated under reduced pressure. Further purification was not necessary. The product is a brownish-yellow, viscous oil (**DQ-N**, Figure 7, 673 mg, 69%). TLC (SiO_2_ TLC; DCM:MeOH:25% NH_4_OH = 10:1:0.01, R_f_ = 0.52;$\;{\left[a\right]}_{20}^{D}$ + 86.7 (c 1.00, CHCl_3_); IR *ν*_max_ 3424, 3294, 2930, 2862, 1621, 1589, 1508, 1474, 1454, 1431, 1356, 1620, 1260, 1229, 1028 cm^−1^; ^1^H NMR (600 MHz, DMSO-d_6_, 25 °C): δ (ppm) 0.76 (1 H, m), 1.35 (1 H, m), 1.41 (1 H, m), 1.49 (1 H, m), 1.67 (1 H, bm), 2.51 (1 H, m), 2.62 (1 H, m), 2.79 (1 H, m), 2.83 (1 H, m), 3.07 (1 H, bm), 3.22 (1 H, bm), 3.32 (1 H, m), 3.94 (3 H, s), 4.64 (1 H, bm), 7.42 (1 H, dd, *J*_1,H,H_ 2.8 Hz, *J*_2,H,H_ 9.2 Hz), 7.59 (1 H, bm), 7.80 (1 H, bm), 7.95 (1 H, d, *J*_H,H_ 9.2 Hz), 8.72 (1 H, d, *J*_H,H_ 4.5 Hz); ^13^C NMR (150 MHz, DMSO-d_6_, 25 °C): δ (ppm) 25.9, 26.0, 26.7, 26.9, 55.7, 57.2, 61.5, 71.2, 79.4, 88.5, 102.9, 120.5, 121.4, 128.7, 129.0, 131.4, 144.3, 147.9 (2 C), 157.1; HRMS-ESI+ *m/z* [M + H^+^] calcd. for C_20_H_24_N_3_O: 322.1919, found: 322.1923.

### 4.3. 3-((3,5-Bis(trifluoromethyl)phenyl)amino)-4-(((S)-((1S,2S,4S,5R)-5-ethynylquinuclidin-2-yl)(6-methoxyquinolin-4-yl)methyl)amino)cyclobut-3-ene-1,2-dione

A solution of **DQ-N** (Figure 7) (200 mg, 0.67 mmol) in chloroform (0.7 mL) was added to a solution of half squaramide (**HSQ**: 322 mg, 0.74 mmol) in chloroform (1.5 mL). This mixture was stirred for 12 h at room temperature. The solvent was removed under reduced pressure, then the crude product was purified by column chromatography on silica gel using DCM: methanol:25% NH_4_OH (10:1:0.01) mixture as an eluent to obtain the pure product as white crystals (**DQ-SQ**, Figure 8, 387 mg, 99%). M.p. 240 °C (decomposes). TLC (SiO_2_ TLC; DCM:MeOH:25% NH_4_OH = 10:1:0.01, R_f_ = 0.52;$\;{\left[a\right]}_{20}^{D}$ −104.8 (c 1.00, CHCl_3_); IR *ν*_max_ 3441, 3315, 3213, 2934, 1796, 1669, 1623, 1573, 1512, 1455, 1380, 1279, 1186, 1129, 1042 cm^−1^; ^1^H NMR (600 MHz, DMSO-d_6_, 25 °C): δ (ppm) 0.76 (1 H, bm), 1.45 (1 H, bm), 1.55 (1 H, bm), 1.65 (1 H, bm), 1.77 (1 H, bm), 2.60 (1 H, m), 2.88 (1 H, m), 2.96 (1 H, bm), 3.28 (1 H, m), 3.34 (1 H, m), 3.57 (1 H, bm), 3.97 (3 H, s), 6.11 (1 H, bm), 7.48 (1 H, dd, *J*_1,H,H_ 2.3 Hz, *J*_2,H,H_ 9.1 Hz), 7.62 (1 H, d, *J*_H,H_ 4.0 Hz), 7.67 (1 H, bm), 7.76 (1 H, bm), 7.97 (2 H, m), 8.01 (1 H, d, *J*_H,H_ 9.1 Hz), 8.39 (1 H, bm), 8.86 (1 H, d, *J*_H,H_ 4.0 Hz), 10.15 (1 H, bs); ^13^C NMR (150 MHz, DMSO-d_6_, 25 °C): δ (ppm) 25.7, 26.3, 26.7 (2 C), 55.1, 55.9, 57.0, 59.0, 71.5, 88.3, 101.6, 115.2, 118.6, 119.3, 122.2, 123.3 (q, *J*_C,F_ 273 Hz), 127.7, 128.9, 129.0, 131.4 (*J*_C,F_ 33 Hz), 131.8, 141.0, 143.2, 144.5, 148.0, 158.0, 162.9, 168.8, 180.4, 185.1; HRMS-ESI+ *m/z* [M+H^+^] calcd. for C_32_H_27_N_4_O_3_F_6_: 629.1987, found: 629.1990.

### 4.4. General Procedure for Michael Addition of Pentane-2,4-dione (**2**) to Trans-β-nitrostyrene (**1**).

To a solution of *trans*-β-nitrostyrene (**1**) (23.4 mg, 0.16 mmol) in the corresponding solvent (1 mL), organocatalyst (5 mol%) was added. Then, pentane-2,4-dione (**2**) (41.5 µL, 40.7 mg, 0.41 mmol) was added to this solution and the resulting reaction mixture was stirred at room temperature. After 24 h, the volatile components were removed under reduced pressure. The crude product was purified by preparative thin-layer chromatography on silica gel using hexane:ethyl acetate 2:1 mixture (R_f_ = 0.36) as eluent to obtain Michael adduct (*S*)-**3** as white crystals. Yields and enantiomeric excess (ee) values can be seen in in the Table 3 and Appendix A. Spectral data are fully consistent with those reported in the literature (the absolute configuration was determined by the optical rotation of the products) [28]. HPLC: Phenomonex Lux Cellulose-3 column (3 mm, 250 × 4.6 mm), eluent CH_3_CN/20 mM NH_4_OAc in H_2_O = 40/60, isocratic mode; 0.6 mL min^−1^; UV detector 222 nm, 5 µL or 10 µL injection, 20 °C. Retention time for (*S*)-**3**: 11.94 min, for (*R*)-**3**: 14.20 min.

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
