# Peer review of "Comparison of Cinchona Catalysts Containing Ethyl or Vinyl or Ethynyl Group at Their Quinuclidine Ring"

_materials, 2019, doi:10.3390/ma12183034_

Round 1

Reviewer 1 Report

This manuscript describes a detailed insight of the cinchona-based organocatalysts based
on the 1,4-addition of 1,3-dioxo compounds to nitroalkenes. As mentioned by authors, such approaches
based on their basicites and substituents on the quinuclidine unit for understanding the catalytic effectivity are rare, providing
researchers to construct new organocatalysts. However, after all, it seems that this manuscript
simply shares with the effectiveness of thiourea or squaramide cinchona derivatives.
In the former part of this manuscript, two possible mechanisms are thinkable on this 1,4-addition.
Based on these results on the manuscript, the reviewer expects to suggest which gets the dominant mechanism.
The reviewer think this discussion is much important for enhancing the contents in the manuscript.
Anyway, the reviewer would recommend that this manuscript is suitable for publication in Materials.

Author Response

Reviewer 1:

’Comments:

This manuscript describes a detailed insight of the cinchona-based organocatalysts based on the 1,4-addition of 1,3-dioxo compounds to nitroalkenes. As mentioned by authors, such approaches based on their basicites and substituents on the quinuclidine unit for understanding the catalytic effectivity are rare, providing researchers to construct new organocatalysts. However, after all, it seems that this manuscript simply shares with the effectiveness of thiourea or squaramide cinchona derivatives. In the former part of this manuscript, two possible mechanisms are thinkable on this 1,4-addition. Based on these results on the manuscript, the reviewer expects to suggest which gets the dominant mechanism. The reviewer think this discussion is much important for enhancing the contents in the manuscript. Anyway, the reviewer would recommend that this manuscript is suitable for publication in Materials.’

Answer:

Thank you for your comments. The DFT calculations performed in this study focuses only on the relevance of the possible tautomers of thiourea derivatives and the comparison of their relative energy levels as well. Based on these results, we cannot decide trustworthily which the dominant mechanism would be. Further experiments and calculations with other methods would be necessary to take sides regarding the possible mechanism.

Reviewer 2 Report

This reviewer can not decide with confidence if the manuscript is important to be accepted, or not.

Author Response

Response to comments made by the Reviewers:

Reviewer 2: This reviewer can not decide with confidence if the manuscript is important to be accepted, or not.

Answer: Thank you for reading our article.

Reviewer 3 Report

It is a very well-written manuscript in which the authors report the preparation of twelve cinchona organocatalysts containing three different susbtituents at the quinuclidine unit, specifically an ethyl, a vinyl or an ethynyl group. All of them were tested in a model Michael addition reaction obtaining good to excellent yields and ee depending on the solvent used. Their studies have been supported by quantum calculations.

The manuscript could be accepted after minor revisions as followings.

On page 4, table 2, 4th column, ‘pKa, NH2’ instead of ‘pKa, NH1’. On page 4, line 129 ‘moiety’ instead of ‘moeity’. On page 5, line 157, in spite of ‘tert-butyl methyl ether’ is perfectly correct, attending to the acronym MTBE is written, maybe it could be helpful writing the solvent name as methyl tert-butyl ether to remark the order of the letters to form MTBE and not TBME, or alternatively, it might be written ‘tert-butyl methyl ether (TBME)’. On page 7, line 193, a blank space between ‘energies’ and ‘can’ should be deleted. On page 14, reference 28, ‘(8),’ should be deleted.

Author Response

Response to comments made by the Reviewers:

Reviewer 3:

‘Comments:

It is a very well-written manuscript in which the authors report the preparation of twelve cinchona organocatalysts containing three different susbtituents at the quinuclidine unit, specifically an ethyl, a vinyl or an ethynyl group. All of them were tested in a model Michael addition reaction obtaining good to excellent yields and ee depending on the solvent used. Their studies have been supported by quantum calculations.

The manuscript could be accepted after minor revisions as followings.

On page 4, table 2, 4th column, ‘pKa, NH2’ instead of ‘pKa, NH1’.’

Answer:

Thank you for your comments. It has been changed accordingly.

‘On page 4, line 129 ‘moiety’ instead of ‘moeity’.’

Answer:

It has been changed accordingly.

‘On page 5, line 157, in spite of ‘tert-butyl methyl ether’ is perfectly correct, attending to the acronym MTBE is written, maybe it could be helpful writing the solvent name as methyl tert-butyl ether to remark the order of the letters to form MTBE and not TBME, or alternatively, it might be written ‘tert-butyl methyl ether (TBME)’.’

Answer:

It has been changed accordingly.

‘On page 7, line 193, a blank space between ‘energies’ and ‘can’ should be deleted.’

Answer:

It has been changed accordingly.

‘On page 14, reference 28, ‘(8),’ should be deleted.’

Answer:

It has been changed accordingly.

Reviewer 4 Report

The authors report a series of cinchona organocatalysts with different substituents at their quinuclidine unit. In particular, they present the synthesis of four organocatalyst classes (hydroxy, amino, squaramide and thiourea). They found large differences in basicity of the quinuclidine unit by measuring the pKa values of twelve catalysts in six different solvents. The effect was examined by testing the catalysts in Michael addition reaction of pentane-2,4-dione to trans-β-nitrostyrene. They found that the 1.6–1.7 pKa deviation in basicity of the quinuclidine unit did not result in significant differences in yields and enantiomeric excesses. The authors also used quantum chemical calculations to investigate the most stable tautomers in the four cases used in this manusript.

My opinion is that this manuscript is interesting, it has been competently done and it is technically correct. Moreover, it is adequate for the readership of this journal. The conclusions are well supported by the results. Therefore, I recommend publication after some revision as follows:

DFT calculations: The authors should clearly state if they have performed vibrational studies to corroborate the minimum nature of the complexes. The basis set used for the optimization is rather low taking into consideration the nowadays power of the computers. Therefore, I recommend the authors to use a triple-zeta basis set with polarization for the calculations. The figures where the different tautomers are shown are quite difficult to understand due to the small size of the models. They should be improved, specially figure 5. In case of the squaramide family of organocatalysts, the authors need to cite several works previously reported by A. Costa and coworkers where they analyze both experimentally and computationally the ability of the squaramido ring to establish H-bonding interactions both as donor and acceptor. In this sense the current manuscript does not properly cite the current literature.

Author Response

Response to comments made by the Reviewers:

Reviewer 4:

’Comments:

The authors report a series of cinchona organocatalysts with different substituents at their quinuclidine unit. In particular, they present the synthesis of four organocatalyst classes (hydroxy, amino, squaramide and thiourea). They found large differences in basicity of the quinuclidine unit by measuring the pKa values of twelve catalysts in six different solvents. The effect was examined by testing the catalysts in Michael addition reaction of pentane-2,4-dione to trans-β-nitrostyrene. They found that the 1.6–1.7 pKa deviation in basicity of the quinuclidine unit did not result in significant differences in yields and enantiomeric excesses. The authors also used quantum chemical calculations to investigate the most stable tautomers in the four cases used in this manusript.

Answer:

Vibrational analysis was performed in the gas phase for the most stable tautomers. These clearly show that these are minima on the potential energy surface. We extended the methods section accordingly: ‘Harmonic vibrational frequencies were computed in the gas phase for the most stable tautomers. These show no imaginary vibrational frequencies, confirming that the computed conformers are minima on the potential energy surface.’

’The basis set used for the optimization is rather low taking into consideration the nowadays power of the computers. Therefore, I recommend the authors to use a triple-zeta basis set with polarization for the calculations.’

Answer:

Originally, for the basis set choice we were limited by the capabilities of the SM8 solvation model, what is implemented only up to 6-31+G** basis set. We switched to the more recent SM12 solvation model, without any limitation on the basis set size and re-computed the energies and the solvation free energies using 6-311++G**. As it is visible in the supporting information, there is small difference between the previous and the new results. Thus, this is an important confirmation about the accuracy of our computations.

’The figures where the different tautomers are shown are quite difficult to understand due to the small size of the models. They should be improved, specially figure 5.’

Answer:

We thank for the reviewer for pointing out this issue. We fully agree and changed the manuscript as follows. Instead of Figure 5 we included the new Figure 4 what contains the chemical formula of the most stable tautomers and the protonated and deprotonated forms. The molecular structures and the coordinates are available in the supplementary material.

’In case of the squaramide family of organocatalysts, the authors need to cite several works previously reported by A. Costa and coworkers where they analyze both experimentally and computationally the ability of the squaramido ring to establish H-bonding interactions both as donor and acceptor. In this sense the current manuscript does not properly cite the current literature.’

Answer:

‘We thank the reviewer for pointing out these papers about the squaramides. We included all the references and highlighted their important contribution as “Electrostatic potential provides important insights into the intermolecular interactions of similar compounds [reference 51 and reference 52].’

Round 2

Reviewer 2 Report

This revised manuscript can be accepted.

Reviewer 4 Report

The authors have revised the manuscript accordingly to my previous recommendations and it is now suitable for publication as it is.